# Automatic prostate and prostate zones segmentation of magnetic resonance images using convolutional neural networks

**Nader Aldoj**[1]                                                        NADER.ALDOJ@CHARITE.DE
[1] *Charite, Department of Radiology, Berlin, Germany*
**Federico Biavati**[1]                                            FEDERICO.BIAVATI@CHARITE.DE
**Miriam Rutz**[1]                                                        MIRIAM@RUTZ-WEB.DE
**Florian Michallek**[1]                                    FLORIAN.MICHALLEK@CHARITE.DE
**Sebastian Stober**[2]                                                          STOBER@OVGU.DE
[2] *Otto-von-Guericke-University Magdeburg, Magdeburg, Germany*
**Marc Dewey**[1]                                                      MARC.DEWEY@CHARITE.DE

**Editors:** Under Review for MIDL 2019

## Abstract

Magnetic resonance imaging (MRI) provides detailed anatomical images of the prostate (PR) and its zones. The importance of segmenting the prostate and the prostate zones, such as the central zone (CZ) and the peripheral zone (PZ) lies in the fact that the diagnostic guidelines differ depending on in which zone the lesion is located. Thus, automatic prostate and prostate zone segmentation from MR images is an important topic for many diagnostic and therapeutic purposes. However, the prostate tissue heterogeneity and the huge varieties of prostate shapes among patients make this task very challenging. Therefore, we propose a new neural network named Dense U-net inspired by the state-of-the-art DenseNet and U-net to automatically segment prostate and prostate zones. It was trained on 141 patient datasets and tested on 47 patient datasets with axial T2-weighted images in four-fold cross-validation manner. The network can successfully segment the gland and its subsequent zones. This Dense U-net compared with the state-of-the-art U-net achieved an average dice score for the whole prostate of $91.2 \pm 0.8\%$ vs. $89.2 \pm 0.8\%$, for CZ of $89.2 \pm 0.2\%$ vs. $87.4 \pm 0.2\%$, and for PZ of $76.4 \pm 0.2\%$ vs. $74.0 \pm 0.2\%$. The experimental results show that the developed Dense U-net was more accurate than the state-of-the-art U-net for prostate and prostate zone segmentation.

**Keywords:** prostate gland, image processing, neural networks, magnetic resonance imaging.

## 1. Introduction

Prostate cancer (PCa) is the second leading cause of death among cancers family in men. Due to the huge increase in prostate screening, PCa is the most commonly diagnosed cancer in American men (Siegel et al., 2016). Accurate prostate segmentation is a very essential step in many medical imaging and image analysis tasks such diagnosis, surgical planning (Wang et al., 2016), quantitative volumetric measurements (Terris and Stamey, 1991) and therapeutic purposes (Sabouri et al., 2017). Therefore, automated segmentation of prostate gland and subsequent zones is of high demand in daily clinical practice. In this work we present a novel network architecture inspired by U-net (Ronneberger et al., 2015) and DenseNet (Huang et al., 2017) and harvests the strength of both networks for the segmentation of prostate gland and its subsequent zones.

## 2. Materials and Methods

In this study, a dataset of 188 patients with axial T2-weighted MR images was used (PROSTATEx challenge). All images were manually segmented and examined by an experienced radiologist. Coarsely and accurately segmented images were included in both training and test sets to examine the hypothesis that the networks can learn the segmentation form the accurately annotated images and correct for the coarsely annotated segmentations. We used 141 patients (including a total of 2927 slices) as the training set and 47 patients (including a total of 980 slices) as the test set where the networks were validated in a four-fold cross-validation fashion. All images were first resampled to a common resolution, normalized and then cropped with a 256x256 pixel window. The developed Dense U-net is based on the U-net architecture with 6 stages in the encoding and decoding part. We replaced the normal stack of convolutional layers with a DenseNet-like architecture which consists of one or two small dense blocks (Figure 1 shows a network with two blocks) separated by transitional layers. Each of the Dense blocks comprises 4 convolutional layers with concatenation connections from all respective previous layers. We tested the Dense U-net with one and two blocks against the classical U-net, and evaluated the segmentation with the manual annotations as the reference standard according to the mean dice score (MDS) $\pm$ 95% confidence interval (CI), standard deviation (Std), median dice score (MeDS), mean relative absolute volume difference (MRAVD) and mean Haussdorf distance (MHD) as a contour consistency measure. Furthermore, visual evaluation of segmentation performance was done where visual dice scores were assigned. Segmentation results (for classical and Dense U-net with two blocks) were inspected in a shuffled and blind manner by an independent radiologist.

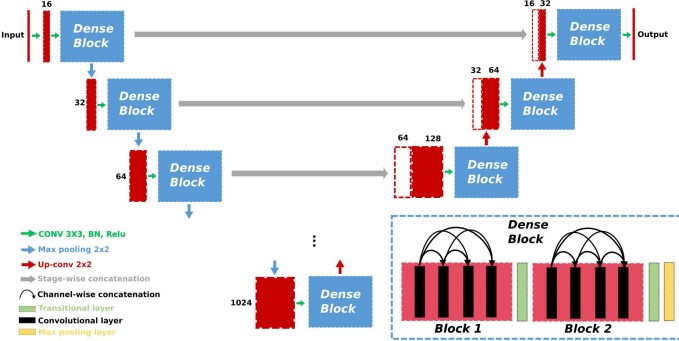

Figure 1: The Dense U-net architecture.

## 3. Results and Discussion

The Dense U-net achieved an average and median dice score for the prostate of 91.2±0.8% and 90.3% with two dense blocks and 89.3±0.8% and 89.1% with one dense block which was higher in comparison to the classical U-net with average and median dice scores of 89.2±0.8% and 88.7%, respectively. In addition, the Dense U-net had a higher dice score of 89.2±0.2% in CZ, and 76.4±0.2% in PZ with two dense blocks and 87.4±0.8% in CZ, and 74.5±0.2% in PZ with one dense block when comparing it to 87.4±0.2% and 74.0±0.2% of the classical U-net. Both networks performed accurate segmentation of the prostate gland and its subsequent zones with details of the statistical measures presented in table 1. Figure 2 shows some segmentation results from the Dense U-net.

Visual dice scores were (mean±standard deviation) 84.4±3.4% for classical U-net, 85.1±3% for Dense U-net, 86.1±3% for human reader in whole-prostate segmentation and 74.3±2.4% for classical U-net, 74.5±2.5% for Dense U-net, 75.8±2.5% for human reader in peripheral zone-only segmentation.

As can be seen in Figure 2B, the network was able to learn an accurate segmentation of the tissue although some of the labels which are used for training were weakly annotated.

The improvement of the performance was due to the nature of the Dense U-net which is based on feature maps concatenation where one convolutional stage has a direct access to all previous feature maps from all subsequent stages and this enables feature maps reuse.

In summary, the developed Dense U-net architecture in both forms was more accurate than the classical U-net for prostate and prostate zone segmentation in terms of Dice score. Although the difference is not statistically significant, it is appreciated by the radiologists.

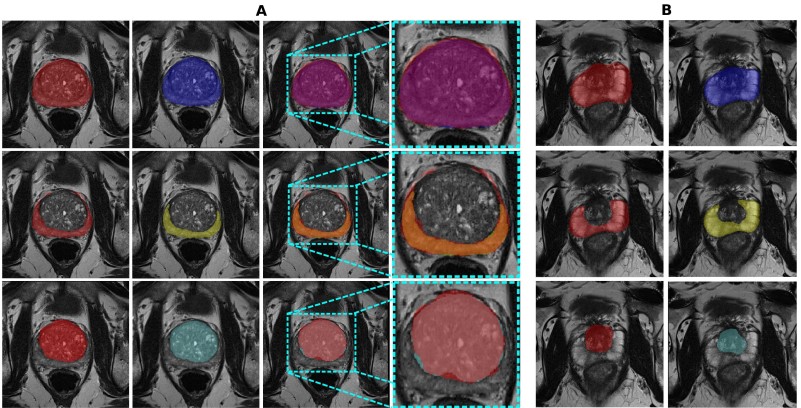

Figure 2: Segmentation results of Dense U-net: A) Columns from left to right show images of the ground truth, predicted mask, the overlap between the two masks and a magnification of the overlap; the rows from top to bottom show images of prostate, PZ and CZ respectively. B) Shows accurate segmentation resulted from Dense U-net (right) in comparison to the weakly annotated labels (left).

Table 1: The statistical measurements of the segmentation results. The number associated with the Dense network refers to the number of dense blocks.

| Network | MDS% | CI 95% | StD(%) | MeDS(%) | MRAVD(%) | MHD (mm) |
|---|---|---|---|---|---|---|
| Classical U-net (PR) | 89.2 | ±0.8 | ±3 | 88.7 | 44.3 | 23.5 |
| Dense-1 U-net (PR) | 89.3 | ±0.8 | ±3 | 89.1 | 45.5 | 23.1 |
| Dense-2 U-net (PR) | 91.2 | ±0.8 | ±3 | 90.3 | 36.1 | 23.3 |
| Classical U-net (CZ) | 87.4 | ±0.2 | ±5 | 86.1 | 15.8 | 14.9 |
| Dense-1 U-net (CZ) | 87.4 | ±0.2 | ±1 | 87.1 | 10.1 | 14.1 |
| Dense-2 U-net (CZ) | 89.2 | ±0.2 | ±5 | 88.1 | 9.6 | 14.2 |
| Classical U-net (PZ) | 74.0 | ±0.2 | ±7 | 75.0 | 21.0 | 17.7 |
| Dense-1 U-net (PZ) | 74.5 | ±0.2 | ±7 | 75.2 | 24.3 | 20.0 |
| Dense-2 U-net (PZ) | 76.4 | ±0.2 | ±7 | 77.2 | 17.2 | 19.9 |

## Acknowledgments

This work is funded by the German research foundation (GRK2260, BIOQIC)

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
