# OpenReview forum: "Automatic prostate and prostate zones segmentation of magnetic resonance images using convolutional neural networks"
_MIDL.io/2019/Conference/Abstract — MIDL Abstract 2019_

### Official Review · AnonReviewer2 · 2019-04-24
**this work mainly shows that a U-Net with or without extensions might be a good choice for image segmentation**

**Rating:** 2
**Confidence:** 3

**Review:**

This abstract combines a U-Net with a DenseNet and evaluates this setup on a prostate zone segmentation task.

The results section is hard to read. It would be better if these would be presented only in tabular form like Table 1, highlighting the best performing in bold. Furthermore, which of these results are significantly better than the baseline. They seem to perform pretty close to the baseline. Which accuracy is required in the clinical practice?
page 2 typo: "learn the segmentation form"
What is meant with the term "Visual dice scores" ?

The abstract is well prepared and the evaluation is reasonable. However, I am not sure if the proposed combination really has any advantages beyond a higher parameter capacity and this work mainly shows that a U-Net with or without extensions might be a good choice for image segmentation.

---

### Official Review · AnonReviewer1 · 2019-04-30
**Segmentation of the prostate and its zones in MR images using a CNN with the architecture that is inspired by U-net and DenseNet.**

**Rating:** 3
**Confidence:** 2

**Review:**

T2-weighted MR scans from the PROSTATEx challenge are used in the experiments. Presented results show that the used network outperforms segmentation with U-net, and that Dense2 U-net outperforms Dense1 U-net, but it would be nice to see whether these differences are statistically significant.

The abstract is overall well written but nevertheless the following could be clarified:
-	Were the manual annotations obtained from the challenge or was it specifically done for this study
-	In the text it is stated that four-fold classification was made but that testing was performed on 47 patients. I assume this is per-fold.
-	What is meant by Visual dice scores and who performed this evaluation?
-	In the Results and Discussion as well as in Fig 2 weak annotations are mentioned. Because this was not mentioned earlier (reference annotations were not described as weak), it is unclear where this comes from and what does this mean.
Very minor, but abbreviations CZ and PZ should also be introduced in the body of the paper.
It would be very nice to in some way indicate whether the achieved performance is sufficient for clinical use (indication of e.g. interobserver performance would be great). Some comparison with existing work would be valuable.

---

### Decision · Program_Chairs · 2019-05-06
**Acceptance Decision**

Accept